# Regulation of Autophagy Machinery in *Magnaporthe oryzae*

**DOI:** 10.3390/ijms23158366

**Published:** 2022-07-28

**Authors:** Nida Asif, Fucheng Lin, Lin Li, Xueming Zhu, Sehar Nawaz

**Affiliations:** 1State Key Laboratory for Managing Biotic and Chemical Treats to the Quality and Safety of Agro-Products, Institute of Biotechnology, Zhejiang University, Hangzhou 310058, China; nidaasif@zju.edu.cn; 2State Key Laboratory for Managing Biotic and Chemical Treats to the Quality and Safety of Agro-Products, Institute of Plant Protection and Microbiology, Zhejiang Academy of Agricultural Sciences, Hangzhou 310021, China; 21616143@zju.edu.cn (L.L.); zhuxm@zaas.ac.cn (X.Z.); 3Center of Agricultural Biochemistry and Biotechnology (CABB), University of Agriculture Faisalabad, Faisalabad 38040, Pakistan; seharnawazpbg@yahoo.com

**Keywords:** *Magnaporthe oryzae*, autophagy, pathogenesis, appressorium, phagophore, autophagosome, autophagy-related techniques

## Abstract

Plant diseases cause substantial loss to crops all over the world, reducing the quality and quantity of agricultural goods significantly. One of the world’s most damaging plant diseases, rice blast poses a substantial threat to global food security. *Magnaporthe oryzae* causes rice blast disease, which challenges world food security by causing substantial damage in rice production annually. Autophagy is an evolutionarily conserved breakdown and recycling system in eukaryotes that regulate homeostasis, stress adaption, and programmed cell death. Recently, new studies found that the autophagy process plays a vital role in the pathogenicity of *M. oryzae* and the regulation mechanisms are gradually clarified. Here we present a brief summary of the recent advances, concentrating on the new findings of autophagy regulation mechanisms and summarize some autophagy-related techniques in rice blast fungus. This review will help readers to better understand the relationship between autophagy and the virulence of plant pathogenic fungi.

## 1. Introduction

Numerous countries rely on agriculture to provide food and a source of income to enhance the economy. In both temperate and tropical agriculture and in the forestry sectors, plant pathogenic fungi inflict substantial economic damage to crops [1]. Rice is one of the most important food crops in the world. About half of the world’s population mainly depends on rice for their livelihood [2]. In the past two decades with the deterioration in global climate, rapid population growth and rampant plant diseases, global rice production has been severely reduced, and food security has been seriously threatened [3]. Therefore, how to ensure food security has become a common concern of all countries in the world. Many fungal diseases inhibit the growth of rice, notably so rice blast disease caused by *Magnaporthe oryzae* [4], which reduces rice production by up to 30% annually around the world [5], which is enough rice for about 60 million people [6]. Fungal infection is a leading factor driving agricultural production shortfalls and in the dramatic growth in widespread lesions in immunosuppressed people [7]. Currently, disease control relies on the use of industrial chemicals such as fungicides [8]. Despite various benefits such as high availability, quick action as well as dependability, fungicides could have a detrimental effect on non-target biological entities because of an extremely toxic and systemic mode of action by disrupting the metabolite levels in the biosynthetic pathway of aromatic amino acids within soil microbes [9]. *M. oryzae* displays profound evolutionary processes throughout pathogenesis, allowing it to enter the cell membrane, construct sophisticated inflammation structures, and ultimately grow within host cells without presenting obvious pathological changes [10]. Throughout all phases of growth, *M. oryzae* attacks the aerial portions of the rice plant, particularly panicles, leaves, stems, and also nodes [11].

Recent studies have shown that the autophagy pathway is a key factor in the production of appressorium turgor and is closely related to the pathogenicity of pathogenic fungi [12,13,14]. Therefore, an in-depth analysis of the autophagy pathway is of great significance for understanding the infection mechanism of the rice blast fungus and for the elaboration of the pathogenic mechanism, thus providing a theoretical basis for the prevention of the pathogenic mechanism in plant pathogenic fungi, especially in the rice blast fungus *M. oryzae*. One will understand how autophagy regulates appressorium development and how to detect autophagy flux in plant pathogenic fungi. Thus, this review will help us better understand the role of autophagy in plant pathogenic fungi.

## 2. Autophagy in *M. oryzae*

Autophagy pathways can be found in yeast, plants, and metazoans [15]. A collection of more than 30 autophagy-related genes (ATGs) is at the center of the autophagic process, each integral to a distinct, yet ongoing stage [16]. There was increased phosphorylation of five autophagy proteins (ATG1, ATG2, ATG3, ATG17, and ATG18) during appressorium formation in *M. oryzae*, but only one site on ATG13 was decreased, suggesting that post-translational ATG alterations are involved in host penetration [17,18]. Recent findings have identified autophagy activation as a key host colonization mechanism to explain how leaf rust pathogens colonize their hosts [19]. The process of autophagy is sometimes induced by a host cell to transport and recycle nutrients received from the host. It can also act as a secondary secretory system [20]. In addition to using pathogen effectors as molecular probes, addressing individual host cargo receptors and adaptors as well as studying autophagy in plant–microbial interactions could provide a deeper understanding of autophagy [21].

In *M. oryzae*, numerous signaling factors have been implicated in appressorium emergence and virulence [22]. *M. oryzae* would be a hemibiotrophic pathogen that goes through a biotrophic phase before transitioning to a necrotrophic phase which stimulates plant cell death [23]. The fungus synthesizes but also secretes a repertoire of effector proteins during the most biotrophic phase of infection that can decrease resistance in plants as well as modify host cell metabolism for the next inflammation [24]. The PKA mitogen-activated protein kinase (MAPK) cascade contributes to recognizing the host surface, while the Pmk1 MAPK pathway is required to develop an appressorium [25,26]. Disseminated dark differentiated structures, which are termed conidium germinate in a water drop on a leaf surface, causing the host plant to become infected [14].

Furthermore, recent research reveals the identification and characterization of possible orthologs of yeast *SNT2* in *M. oryzae*, termed *MoSNT2*. A *MoTor* pathway regulates mo*SNT2* expression to cause autophagy and rice infection through *MoTor* signaling. It has been shown that *MoSNT2* regulates infection-associated autophagy and plant infection by the rice blast fungus through its epigenetic association with *MoTor* signaling [27]. Researchers also discovered that HAT Gcn5 was an important regulator of *M. oryzae* conidiation and pathogenicity. Comparative acetylome and transcriptome analyses revealed a set of histone and non-histone proteins that could be *M. oryzae* GCN5-mediated acetylators [28]. Moreover, metabolic forms of host manipulation by the fungi were also highlighted in a recent study. An essential component of pathogenicity is the absorption of 12OH-JA through hydroxylated jasmonic acid, which is produced by monooxygenase [29]. Recently Meng and his coworker worked on the activation of autophagy in *M. oryzae* and determined that *MoWhi2* is specifically required for mitophagy and the invasive growth of hyphae. These results ultimately showed that *MoWhi2* contributes to a reduction in conidiation and virulence in *M. oryzae* mutants [30].

### Morphology and Physiology of Appressorium

It would be useful to understand whether *M. oryzae* reacts to starvation stress and how its gene expression changes in response to changes in nutritional status induced by fast external changes to provide insight into the genetic control of plant infection [31]. The MPG1 hydrophobic gene is specifically expressed during appressorium development [32]. When the germ tube apex detects the hydrophobic environment of the target area, it divides into the appressorium [33], a highly melanized and domed-shaped infecting layout [34]. *M. oryzae* may provide an important clue as to how plants respond to starvation stress and modulate gene expression in response to rapid changes in external nutrient status [35]. An appressorium arises in the absence of nutrients and understanding how it develops may provide insight into the genetic control of plant infection [36]. In addition to its putative role in conidiation, normal appressorium development, and invasive growth, SPM1 encodes a putative subtilisin serine protease [37].

Appressorial autophagy utilizes cellular constituents retrieved from a conidial cell that may promote appressorium development whereas conidial autophagy results in the breakdown of conidial components along with apoptotic cells [38,39]. The appressorium generates strong hydrostatic turgor pressure throughout morphogenesis that induces a thin spike immediately penetrating into the host epidermis; coinciding together, the germination conidium as well as the growing appressorium’s autophagy-association induces apoptosis [40]. Appressoria are generally apparent as highly distinct structures at the terminals from the germ tubes however, they might be hard to determine morphologically in certain circumstances. They may originate from the hyphae under critical conditions [41]. The fungus generates spherical hyphae that are encapsulated inside a cell membrane after penetrating plant tissues and colonizing the host plant that forms a fungal sore on which it starts sporulation [42]. Once an appressorium is divided, the germ tube and spore go through autophagy and nuclear destruction and lack a cytoplasm, just like in *M. oryzae* [43,44] (Figure 1).

## 3. Mechanism of Initiation and Inhibition of the Autophagic Process

Autophagy, which means self-eating, is one of the key cellular processes responsible for cell destruction. Macro-autophagy is the most recognized and would be now referred to as autophagy [45]. Autophagy has been proposed in plants to contribution in the repurposing of proteins and metabolites as well as in many physiological activities under starvation conditions [46]. G-protein/cAMP signaling is essential for the detection of host surface stimulation and invasion into host tissue in *M. oryzae*. An autophagosome is a double-membrane membrane formed by the phagophore sequestering cytoplasmic elements. All forms of autophagy, including glycolysis, use the autophagosome- a molecular apparatus that participates in the creation of sequestering vesicles as a fundamental process [47]. The process starts in nutritional deficiency (starvation), usually carbon, sucrose and nitrogen [48].

Autophagy in fungi can be initiated through food deprivation or rapamycin therapy [49], resulting in the suppression of mammalian target of rapamycin (mTOR) kinase activity and ultimately activates the autophagy [50,51]. Following autophagy induction, the first phagophore is synthesized at the phagophore assembly site (PAS) and is engulfed in the double-membraned autophagosome when phagophore extension and curving occur [14]. A fusion of an autophagosome and an endocytic and lysosomal compartment forms an autolysosome [52]. Autolysosomes disintegrate once autophagy charges have been degraded, contributing to the regeneration of the lysosomal pool via autophagic lysosome reformation, and contributing to the regeneration of autophagy charges. The discharge of the autophagic body, which is encompassed by the internal autophagosomal membrane, is caused by the integration outer surface of the autophagosome with the vacuolar membrane (Figure 2). In addition, the Cvt process was confirmed as involved in the autophagy pathway in nutritionally rich situations. This Cvt pathway is indeed an autophagy-related mechanism that functions during growing environments and performs a biosynthetic contribution by bringing hydrolytic enzymes such as amino-peptidase I (PrApe1), towards the yeast vacuole, as shown in Figure 2.

## 4. Selective Autophagy

Selective autophagy is a eukaryotic specific autophagy pathway that selectively degrades specific proteins or organelles, and it’s ubiquitous in plants, animals and fungi [53]. By using selective autophagy, organelles and proteins aggregates, damaged or obsolete organelles, and other biological components can be removed [54]. Recently, new studies found that selective autophagy play vital roles in the cell development and pathogenicity in pathogenic fungi. Here we discuss the principal mechanisms behind selective autophagy in plant pathogenic fungi, with a focus on pexophagy and reticulophagy [55], which has been the best-described type of selective autophagy to date [56,57].

### 4.1. Pexophagy

Pexophagy refers to the selective removal of peroxisomes. The primary purpose of pexophagy is to remove extraneous organelles from various fungi [58]. The number, size, and function of peroxisomes depend on the cellular type and metabolic needs. They are also involved in intracellular signaling pathways, including redox, lipid, inflammatory, and immune signaling. In response to changing environmental conditions, peroxisome homeostasis is maintained by balancing biogenesis and degradation [59]. Under various environmental and cellular stress conditions, pexophagy maintains both organelle integrity and number, which are essential to cellular homeostasis [60]. Byproducts of oxidative reactions include reactive oxygen and nitrogen species, and removing damaged organelles maintains a balanced redox state in the cell.

As a result of our earlier studies and research, we hypothesized that *M. oryzae* secretes effectors to rice cells located in peroxisomes, which are responsible for regulating rice immune responses [61]. Peroxisomes of plants are found to express MoPtep1. The expression of MoPtep1 is significantly increased during an infection with *M. oryzae* [62]. Evolutionarily, MoPtep1 was conserved in plant pathogenic fungi, according to sequence analysis. *M. oryzae* was found to be pathogenic when MoPtep1 gene knockouts were performed. Peroxisomal turnover is regulated by pexophagy in old tissues damaged by high levels of hydrogen peroxide, under both favorable and stressful conditions, thereby maintaining the quality and quantity of peroxisomes [63,64]. Naqvi and his research fellows created a deletion of mutants of potential pexophagic-specific genes and rearranged them in accordance with conidiation and the pathogenic properties on the basis of identification of Pex14 as an essential protein for asexual virulence in *M. oryzae* [65]. Further results of the research indicate that the proteins present in *M. oryzae* such as *MoSnx41* likely perform a function of Snx42/Atg20 and Snx41 [66].

### 4.2. Reticulophagy

Synthesis and folding of membrane and secretory proteins take place in the ER. In 2005, the concept of ER-phagy was introduced; however its mechanism is still ambiguous. It is important to maintain homeostasis of the internal environment of the ER to prevent unfolded proteins and misfolded proteins from accumulating in the ER cavity and causing retention [67,68]. This will induce ER stress, and UPR, which are prone to triggering the overall remodeling of the ER; so, maintaining the homeostasis of the internal environment of the ER is important [69]. Most secretory and membrane proteins in eukaryotic cells are molecularly modified and folded through the use of numerous molecular chaperones in the traditional ER Golgi secretory pathway [70] to deliver apoplastic effectors, with exocyst complexes to deliver cytoplasmic effectors [71]. ER stress, resulting from accumulated misfolded or unfolded proteins in the ER [72], is associated with autophagy, which is selective degradation of the ER by the cell [56]. Specifically, *M. oryzae* researchers have focused on an enzyme that regulates autophagy pathway by studying the homeostasis of sterols synthesized in the endoplasmic reticulum [73]. The results of this study have shown that MoVast1 regulates conidiation and pathogenicity of rice blast fungus [74]. According to the proposed study, loss of MoVAST1 leads to conidiostatic defects andan impaired appressorium, which subsequently reduce pathogenicity [75].

### 4.3. Assessment of Autophagy-Related Techniques

In-vitro techniques:

Autophagy was assessed and recorded using a variety of *in-vitro* technologies [76]. Their potential application in plant autophagy investigations would be explored.

(a)*Electron Microscopy:* EM was one of the first approaches used to define autophagy and remains among the most efficient ways for monitoring autophagy in cellular membranes and for quantification of autophagic accumulation. However, for interpreting EM one must have specialized knowledge because there are various considerations for precisely describing autophagosomes and autolysosomes [77]. EM could discriminate between a particular and pathological morphology, including inadequate fusion of the autophagosome and lysosome, dependent on the condition of the related cell. In addition to revealing the morphology of autophagic structures with a resolution within the nm range, it shows these structures in their natural environment and identifies them exactly among the rest of the cellular components.(b)*Molecular Markers:* Proteins involved in autophagy, specifically those that were destroyed by autophagy, were utilized to detect autophagic activation. Many of these are currently used in plants. Plants that have been knocked out or modified for such markers are immensely helpful for studying autophagy-related traits in various experiments [78]. Because of its permanent connection with pre-autophagosomal components, the Atg8 family protein is widely being used as an efficient and comprehensive cytogenetic marker for autophagy detection. An increase in LC3-II amount in the presence of the inhibitor is generally indicative of flux, so the analysis is done in the absence and presence of lysosomal protease or fusion inhibitors.(c)*Long-Lived Protein Degradation:* As autophagy seems to be comprised of the breakdown of long-lived proteins, measuring its rotation appears to be an effective technique of monitoring autophagy rates in cells. Using metabolic labeling, the breakdown of all long-lived proteins can be monitored within the general approach [79]. To mark the newly generated proteins, a highly radioactive tagged amino acid such as valine or leucine, could be used.(d)*Selective Autophagic Degradation of Proteins*: While it is thought that autophagy is a nonselective process, certain proteins tend to be destroyed preferentially because of autophagy. Using autophagy, a green florescent protein (GFP) or DsRed complex target to the chloroplast as well as a GFP fusion of rubisco are delivered towards to vacuole [80]. Rubisco more-or-less receives the mains part of the plant’s nitrogen that participates in carbon fixation in chloroplasts. It is secreted from the chloroplast components known as rubisco-containing bodies (RCBs) which can supply nitrogen from leaves to other tissues [81]. RCB appears to coincide in autophagic compartments, implying that rubisco is absorbed by autophagosomes and then transported towards the vacuole. This mechanism is reliant on ATG genes, emphasizing its transporting autophagic nature.(e)*Tests of Mitochondrial Autophagy (Mitophagy):* As it has been considered that autophagy is a basic method regulating the characteristics of organelles, mitochondrial bodies are frequent victims of autophagic destruction [82]. Mitophagy describes the specific destruction of mitochondria that undergo the process of autophagy [83]. A strategy for detecting mitophagy in yeast has been recently discovered. The process relies on the use of a GFP-tagged mitochondrial protein and indeed the monitoring of vesicle emissions of the green fluorescent protein followed by chimeral disintegration [84].

## 5. Discussion and Future Directions

Autophagy is the key process that governs cellular life and death, and it is closely regulated by complex molecular pathways [85]. Autophagy is believed to be an important regulatory mechanism in both fungal infection and plant resistance that help to renovate cell cytosol by removing dead components, even though it is questionable if autophagy proves predominantly a cell survival reflex or a part of the PCD process of cells that are simultaneously suffering hypersensitive response (HR) [86]. This will be significant to identify how autophagy in plenty of other fungal species is confined to the conidium suffering apoptosis or either appears in the appressorium [87]. As a major component of plant immunity and fungal infection-related development, autophagy has a very important role in both fungal pathogenesis and disease resistance. In relation to cytosolic components, damage organelles, selective autophagy has been shown to be a highly regulated and specific degradation pathway in the body [88]. Researchers studying the pathogenicity mechanism of *M. oryzae* have helped them discover new methods to control rice blast, including improving farming practices, breeding disease-resistant varieties, and developing new antifungal medications [89]. It often triggers extra arbitrary effects, such as cell death propagation, when applied to plant microbial interactions with Atg variants [90,91]. A pathogenic bacterium is delivered during the use of these variants, which in turn triggers numerous immune responses [92]. Selective autophagy is best-characterized with the Cvt pathway. This serves as a model for how autophagic machinery delivers specific cargoes to the vacuole [93]. There are severe disorders of peroxisome biogenesis caused by a lack of functional peroxisomes or by single peroxisome enzyme deficiencies, which illustrate the importance of peroxisomal metabolism [94]. It is important to investigate the regulatory mechanisms of and diverse developmental implications for the endoplasmic reticulum and peroxisome dynamics during the autophagic process in fungi.

In plant pathogenic fungi, autophagy is an essential pathway for survival and virulence. However, it is still unknown how autophagy originates, what factors influence the process on functionality, where the autophagosome membrane originates from, and why a specialized type of autophagy influences pathogenic potential in certain filamentous fungi. In addition, the detection method of autophagy is limited in fungi. In this review, we summarized the roles of autophagy in detail in plant pathogenic fungi and introduced some methods for autophagy flux detection based on recent new research. As a result of recent studies, new details are emerging regarding the processes involved in the invasion of tissues by *M. oryzae*, as well as the effectors needed for the suppression of plant immunity and proliferation of fungi. There is a need to investigate the specific molecular roles of *M. oryzae*, as this could potentially allow testing using both protein and genetic variations. It is, however, necessary to do further research in order to better understand the mechanisms and activities of autophagy in fungi.

## Figures and Tables

**Figure 1 ijms-23-08366-f001:**
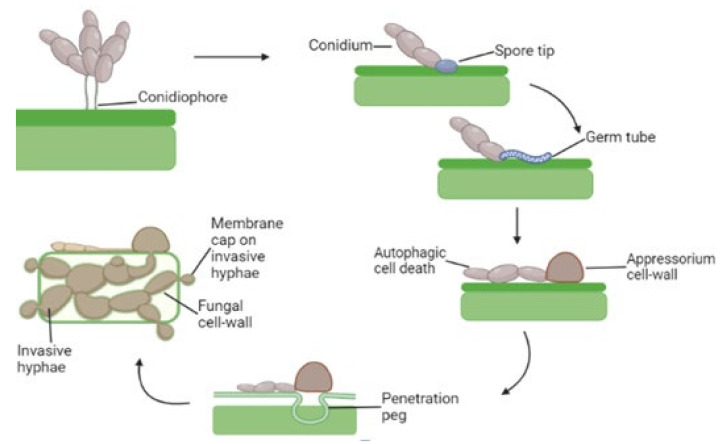
Life cycle of *M. oryzae*. An appressorium is developed from a conidium that produces a penetration peg. This leads to invasive hyphae growth in the host cell. During the infection process of *M. oryzae*, a large number of conidia adhere to the surface of the host leaf through the viscous liquid secreted from the tip and then the appressorium is formed on the tip of the germ tube after sensing the hydrophobic environment on the lead surface. During the process of appressorium development, the material in the conidia is degraded by the autophagy through huge turgor pressure (up to 8 MPa). Under such a huge pressure, the infection pigment formed at the base of the appressorium relies on strong mechanical pressure to pierce the epidermal cells of the host to achieve the purpose of infection.

**Figure 2 ijms-23-08366-f002:**
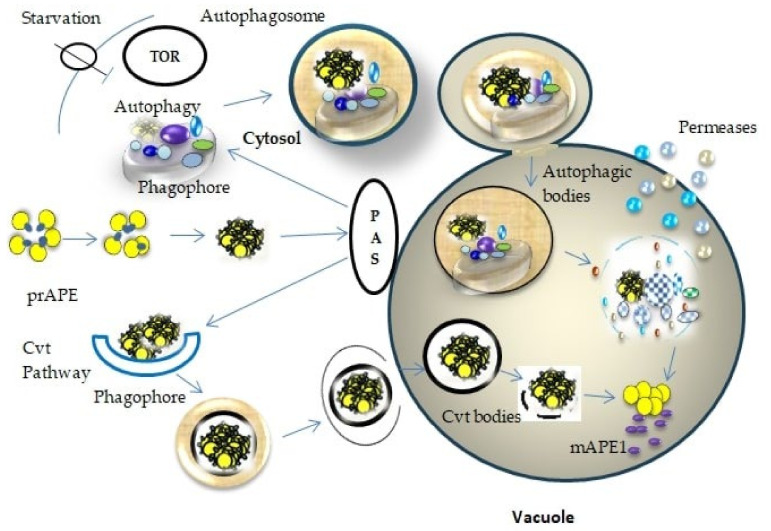
Schematic diagram of the autophagosome at a cellular level. Once the autophagy started induction, the phagophore assembled at PAS. The autophagosomal bodies fuse with the vacuole, releasing the autophagic bodies. These bodies are degraded by hydrolytic enzymes and exported into the cytoplasm for reuse.

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
