# Peer review of "Regulation of Autophagy Machinery in Magnaporthe oryzae"

_ijms, 2022, doi:10.3390/ijms23158366_

Round 1

Reviewer 1 Report

Comments and Suggestions for Authors

Regarding the manuscript “Regulation of autophagy machinery in Magnaporthe oryzae” submitted for consideration by IJMS.

The review seems to be a succession of ideas taken from published papers. In the same paragraph 3 references are mentioned (3-5).  E.g. Many fungal diseases inhibit the growth of rice, notably the rice blast disease caused by Magnaporthe oryzae [3], which reduces rice production by up to 30% annually in the world [4], enough rice for about 60 million people [5]. I recommend the authors to demonstrate that the same reference has been cited several times in the context that it contains more information regarding the review.

“The fungus generates spherical hyphae which are encapsulated inside a cell membrane after penetrating plant tissues and colonizing the host plant that forms a fungal sore  on which it starts sporulation as shown in figure 1 [49]”. The reference 49 does not contain figure 1 which seems to belong to the authors but they lost appressorium formation and maturation as well as autophagic cell death. There is no information about the fact that appressorium becomes melanized and develops substantial turgor. If it make sense for review Life cycle has to be entire.

There is no correlation between successive paragraphs. E.g. Once an appressorium is divided, 130 the germ tube and spore go through autophagy and nuclear destruction and lack cyto- plasm, just like in M. oryzae [45] [50]. Up till now more than 30 autophagy-related genes (ATG) are discovered in yeast and have been found to be conserved in most eukaryotic cells. During pathogenic incursion seems to be a cause of inflammation in multicellular organisms that is accountable for lysosomal degradation of proteins and organelles [51].

Is figure related 2 to M. oryzae or not; as rows 150-153  there is not reference mentioned.

“In 1993, the majority of the ATG genes essential for autophagosome production in yeast were discovered [57] [58]”- !!!

!!!57= Zheng, H., et al., Small GTPase Rab7-mediated FgAtg9 trafficking is essential for autophagy-dependent development and pathogenicity in  Fusarium graminearum. PLoS genetics, 2018. 14(7): p. e1007546.

!!!58 =  Chung, T., A.R. Phillips, and R.D. Vierstra, ATG8 lipidation and ATG8mediated autophagy in Arabidopsis require ATG12 expressed  from the differentially controlled ATG12A AND ATG12B loci. The Plant Journal, 2010. 62(3): p. 483-493.

Rows 199-205 By using selective autophagy, organelles and proteins aggregates, damaged or obsolete organelles, and other biological components can be removed [59]. Plants, yeast, and  mammalian systems have all shown selective autophagy of several organelles, here we discuss the principal mechanisms of selective autophagy in both yeast and mammals, (????) with  a focus on pexophagy and reticulophagy [60], which has been the best-described type of  selective autophagy to date [61]. The core autophagic machinery and selectivity factors of  post selective autophagy share a common mechanism [62] –which is common mechanism? It is not clear If 4. Selective Autophagy is regarding to plants, yeast, and  mammalian systems or mostly to Magnaporthe oryzae.

Conclusion and Future directions should be their own in the light of the current state of research but in reality they include references

My main criticism of this manuscript is that it is too short and often the different sub-sections feel somehow completely disconnected.

09.06.2022

Comments and Suggestions for Authors

Regarding the manuscript “Regulation of autophagy machinery in Magnaporthe oryzae” submitted for consideration by IJMS.

The review seems to be a succession of ideas taken from published papers. In the same paragraph 3 references are mentioned (3-5).  E.g. Many fungal diseases inhibit the growth of rice, notably the rice blast disease caused by Magnaporthe oryzae [3], which reduces rice production by up to 30% annually in the world [4], enough rice for about 60 million people [5]. I recommend the authors to demonstrate that the same reference has been cited several times in the context that it contains more information regarding the review.

“The fungus generates spherical hyphae which are encapsulated inside a cell membrane after penetrating plant tissues and colonizing the host plant that forms a fungal sore  on which it starts sporulation as shown in figure 1 [49]”. The reference 49 does not contain figure 1 which seems to belong to the authors but they lost appressorium formation and maturation as well as autophagic cell death. There is no information about the fact that appressorium becomes melanized and develops substantial turgor. If it make sense for review Life cycle has to be entire.

There is no correlation between successive paragraphs. E.g. Once an appressorium is divided, 130 the germ tube and spore go through autophagy and nuclear destruction and lack cyto- plasm, just like in M. oryzae [45] [50]. Up till now more than 30 autophagy-related genes (ATG) are discovered in yeast and have been found to be conserved in most eukaryotic cells. During pathogenic incursion seems to be a cause of inflammation in multicellular organisms that is accountable for lysosomal degradation of proteins and organelles [51].

Is figure related 2 to M. oryzae or not; as rows 150-153  there is not reference mentioned.

“In 1993, the majority of the ATG genes essential for autophagosome production in yeast were discovered [57] [58]”- !!!

!!!57= Zheng, H., et al., Small GTPase Rab7-mediated FgAtg9 trafficking is essential for autophagy-dependent development and pathogenicity in  Fusarium graminearum. PLoS genetics, 2018. 14(7): p. e1007546.

!!!58 =  Chung, T., A.R. Phillips, and R.D. Vierstra, ATG8 lipidation and ATG8mediated autophagy in Arabidopsis require ATG12 expressed  from the differentially controlled ATG12A AND ATG12B loci. The Plant Journal, 2010. 62(3): p. 483-493.

Rows 199-205 By using selective autophagy, organelles and proteins aggregates, damaged or obsolete organelles, and other biological components can be removed [59]. Plants, yeast, and  mammalian systems have all shown selective autophagy of several organelles, here we discuss the principal mechanisms of selective autophagy in both yeast and mammals, (????) with  a focus on pexophagy and reticulophagy [60], which has been the best-described type of  selective autophagy to date [61]. The core autophagic machinery and selectivity factors of  post selective autophagy share a common mechanism [62] –which is common mechanism? It is not clear If 4. Selective Autophagy is regarding to plants, yeast, and  mammalian systems or mostly to Magnaporthe oryzae.

Conclusion and Future directions should be their own in the light of the current state of research but in reality they include references

My main criticism of this manuscript is that it is too short and often the different sub-sections feel somehow completely disconnected.

09.06.2022

Reviewer 2 Report

The review defines a significant gap in knowledge..The manuscript is clear, relevant for the field.

The structure and organization of paragraphs could be improved.

Figures are appropriate, conclusions are consistent.

Conclusions are supported by the citations

Reviewer 3 Report

Manuscript Number: ijms-1764638

Title: Regulation of autophagy machinery in Magnaporthe oryzae

The manuscript presents a review of the literature on the autophagy mechanism of Magnaporthe oryza, a species causing rice blast disease - the most important plant fungal disease in the world. It is worth to emphasize that M. oryzae is perceived as a model organism for studying pathogen infections and interactions between plants and fungi. The authors briefly described the latest reports on the regulation of autophagy mechanisms and autophagy-related techniques in the rice blast fungus. The review is based on a large amount of literature mainly from recent years, which indicates that the subject is up-to-date and is of great interest to the scientific community. The possibility of finding information in one article can be very helpful for people potentially interested in reducing losses in agricultural production, especially in the context of food shortage in the world caused by the geopolitical situation that has been dynamically changing recently. The paper is quite fairly written and organized however there are some minor points that need to be improved. Below, there are some comments:

1)    Line 59: “Several pathogens and filamentous diseases, as well as some other pathogens…”

It's a bit of an unfortunate formulation where, in which "some other pathogens" suggest that specific pathogens have been named before, but instead there is expression "some pathogens". It is advisable to re-edit this part of sentence.

2)    Lines 79-80 and 81-82: Sentences: “M. oryzae is a hemibiotroph fungus that forms a biotrophic…” and “M. oryzae would be a hemibiotrophic pathogen that goes through a biotrophic …”

One of the sentences is redundant as both contain similar information.

3)    Line 213: “homeostasis is maintained by pexophagy by maintaining” the author also should try to rewrite this part of sentence.

Round 2

Reviewer 1 Report

I recommend the authors to include in future review more information from the studied bibliography